# Joint Light-Sensitive Balanced Butterfly Optimizer for Solving the NLO and NCO Problems of WSN for Environmental Monitoring

**DOI:** 10.3390/biomimetics8050393

**Published:** 2023-08-26

**Authors:** Fei Xia, Ming Yang, Mengjian Zhang, Jing Zhang

**Affiliations:** 1Electrical Engineering College, Guizhou University, Guiyang 550025, China; fxia@gzu.edu.cn (F.X.); zhangjing@gzu.edu.cn (J.Z.); 2School of Computer Science and Engineering, South China University of Technology, Guangzhou 510006, China

**Keywords:** balanced butterfly optimizer, bio-inspired optimization, wireless sensor network, node localization, node coverage, environmental monitoring

## Abstract

Existing swarm intelligence (SI) optimization algorithms applied to node localization optimization (NLO) and node coverage optimization (NCO) problems have low accuracy. In this study, a novel balanced butterfly optimizer (BBO) is proposed which comprehensively considers that butterflies in nature have both smell-sensitive and light-sensitive characteristics. These smell-sensitive and light-sensitive characteristics are used for the global and local search strategies of the proposed algorithm, respectively. Notably, the value of individuals’ smell-sensitive characteristic is generally positive, which is a point that cannot be ignored. The performance of the proposed BBO is verified by twenty-three benchmark functions and compared to other state-of-the-art (SOTA) SI algorithms, including particle swarm optimization (PSO), differential evolution (DE), grey wolf optimizer (GWO), artificial butterfly optimization (ABO), butterfly optimization algorithm (BOA), Harris hawk optimization (HHO), and aquila optimizer (AO). The results demonstrate that the proposed BBO has better performance with the global search ability and strong stability. In addition, the BBO algorithm is used to address NLO and NCO problems in wireless sensor networks (WSNs) used in environmental monitoring, obtaining good results.

## 1. Introduction

Sensor networks play a vital role in the study of intelligent environmental monitoring systems [1,2]. Node localization optimization (NLO) and node coverage optimization (NCO) are important problems in WSNs [3,4,5,6], which are the core component of the Internet of Things (IoT) for intelligent management. The emergence of SI optimization algorithms provides novel approaches for many practical optimization problems that are difficult to solve. At present, the global positioning system (GPS) and Beidou navigation system of China can accurately locate the target; however, certain places, such as indoor locales, dense forests, mountains, caves, and underwater situations, satellite positioning technology cannot precisely locate the target due to the interference or even shielding of the signal by the obstruction. Thus, there is great significance involved in studying the NLO and NCO problems in the context of WSNs, along with substantial challenges. Recently, swarm intelligence (SI) optimization algorithms have been widely used in WSN node deployment, positioning, routing, and other problems [7,8,9]. It is worth noting that several heuristic algorithms have been proposed based on the social behavior and physiological characteristics of butterflies.

There are a great many species of butterflies [10,11,12], most of which rely on scent perception for foraging and mating; others have strong visual sensitivity, and rely on their vision for foraging [13]. In past studies, researchers have been inspired by the inherent characteristics of butterflies, including behaviors such as migration, flight, foraging, and mating. Several swarm intelligence algorithms have been inspired by these behaviors of butterflies, including monarch butterfly optimization (MBO) [14], artificial butterfly optimization (ABO) [15], butterfly optimization algorithm (BOA) [16], and more. MBO is an optimization algorithm proposed by designing a search strategy based on the migration characteristics of monarch butterflies. The ABO algorithm designs the butterfly’s flight strategy solely from the artificial point of view, then divides the butterfly population into two categories and updates their positions during flight using the relative position. The BOA established an odor foraging model of butterflies; however, it cannot solve all 23 CEC benchmark functions [17,18]. The reason for this is that the odor model generates negative numbers for certain problems, which biases the search range of individuals and leads to optimization with imaginary numbers for certain benchmark functions.

Aiming at the shortcomings of BOA-based modeling and inspired by the literature [19], a novel smell- and light-sensitive balanced butterfly optimizer (BBO) is proposed in this paper, distinguished by a simple structure and conforming to the characteristics of actual butterfly behaviors. The global and local search strategies of the BBO are inspired by the natural characteristics of different species of butterflies. First, different species of butterflies have light-sensing characteristics, which should be considered because their visual function is better than their odor function. Second, the smell perception of butterflies is used in the process of foraging or courtship, and the corresponding odor value from the mathematical point of view is positive. Therefore, the absolute value of smell characteristics is introduced to ensure the rationality of the proposed BBO algorithm. The main contributions of this study are summarized from the above motivation as follows:According to the smell perception and light perception characteristics of actual butterflies in nature, a novel balanced butterfly optimizer (BBO) is proposed. Inspired by the physical properties of individuals, the search strategy is designed to avoid falling into local optima.The superiority of the BBO is verified by numerical optimization experiments on all 23 CEC benchmark functions, then the results are compared with a variety of advanced SI algorithms.The NLO and NCO problems of WSNs in IoT contexts are modeled; the proposed BBO is used to address these problems, showing better performance than several well-known SI algorithms.

## 2. Related Works

Among the existing swarm intelligence algorithms, the Firefly algorithm (FA) [20] is a typical SI method inspired by the photosensitivity characteristic of fireflies in nature. The fruitfly optimization algorithm (FOA) [21] is another typical SI algorithm that uses the social behavior of fruit fly odorant foraging and updates individual positions by smell. In addition, FA [22] and FOA [23] have been applied to the NLO problem of WSNs. Moreover, it is worth noting that the classic intelligent algorithms include GA [24], PSO [25], DE [26], FA [20] and GWO [27], among others [28,29,30,31]. DV-Hop localization is a classic non-ranging positioning algorithm; its principle is a distributed positioning method via distance vector hopping and positioning. Kanwar et al. [32] proposed an optimized DV-Hop localization method for sensor node displacement in WSNs using the PSO algorithm. Ouyang et al. [33] proposed an improved GA to optimize the node DV-Hop localization optimization algorithm for the NLO issue. However, the location accuracy of the above methods needs to be further enhanced.

Traditional WSN node localization technologies are mainly categorized into range-based location [34] and range-free location approaches [35]. The typical range-based location approaches use the received signal strength indicator (RSSI) [36], the time of arrival/difference of arrival (TOA/TDOA) [37,38], the angle of arrival (AOA) [39], etc. The typical range-free location approaches include centroid localization (EL), weighted centroid localization (WCL) [40], DV-Hop [32], multidimensional scaling (MDS) localization [41], approximate point-in-triangulation (APIT) localization [42], etc. It is worth noting that the range-based location approaches have higher positioning accuracy than those without ranging, while those without ranging have lower cost, lower power consumption, stronger resistance to measurement noise, and simpler hardware equipment. It is particularly important to note that as long as a range-free location alogorithm can meet the accuracy needs of the application it is generally preferable. A brief summary of NLO and NCO problems using SI algorithms is detailed in Table 1.

DV-Hop is a distributed localization algorithm that uses the routing and positioning of the distance vectors of nodes. It is popular due to its simplicity and low equipment requirements. SI algorithms have proven particularly effective when applied to the optimization of traditional localization algorithms. Shi et al. [43] proposed a modified PSO-DV-Hop method and used a path-matching strategy for searching the shortest path between anchor nodes and independently determining the distance of the average hop from unknown nodes to target anchors. Han et al. [44] used the DE algorithm combined with a weighted DV-Hop method in which the second step of the weighted DV-Hop algorithm used the average hop distance, then the location of unknown nodes was optimized by the DE. Zhang et al. [45] used an enhanced sparrow search algorithm (SSA) to optimize the DV-Hop localization method through a multi-hybrid strategy to improve the SSA and a dual communication radius method to modify the minimum number of hops between nodes while reducing the estimated distance error. In [46], a modified DV-Hop was proposed with an enhanced squirrel search algorithm, which was utilized to estimate the distance from unknown nodes to anchor nodes.

The optimization of node coverage plays a crucial role in improving the capabilities of the WSN work area. Notably, SI algorithms have played a significant role in the NCO problem for WSNs. Yang et al. [47] solved the sensor coverage problem with improved FA taking into account the target coverage and network node connectivity. A resampled PSO method was proposed by Wang et al. [48], which was utilized to optimize the node coverage control of WSNs in the IoT context. There are two optimization algorithms inspired by wolf packs which have been used to address the NCO issue. The first is the GWO-EH algorithm [49] and the other is the wolf pack algorithm (WPA) [50] coupled to the coverage-oriented method. Zhang et al. [51] proposed a node coverage optimization method via the hybrid HPSBA using different simulation areas. Quite successful results have been achieved by the aforementioned SI approaches; however, avoiding local optima remains difficult for a number of challenging problems, making the study of novel heuristic algorithms both necessary and significant.

The above summary describes the use of different kinds of SI algorithms to address the NLO and NCO problems of WSNs. Regardless of whether two-dimensional or three-dimensional positioning is used for the NLO, shortcomings such as insufficient positioning accuracy and low positioning efficiency are encountered. This study proposes the novel BBO algorithm combined with DV-Hop and applies it to the WSN node localization problem. In addition, the proposed BBO algorithm is used to address the NCO problem in WSNs to modify the node coverage ratio. Notably, the proposed BBO can be used to solve other optimization problems as well.

## 3. Problem Descriptions

### 3.1. Node Localization Optimization (NLO) Problem in WSNs

Anchors and unknown nodes are crucial for the NLO problem in WSNs. Anchor nodes are installed with GPS positioning devices, and their coordinates can be known through satellite positioning. The cost and energy consumption of anchor nodes is usually much higher than that of ordinary sensor nodes. The localization method is to locate and optimize the position of an unknown node through an anchor node with a known location. The description of the NLO problem is as follows: (1) there are *m* anchors with known positions and *n* unknown nodes with undetermined positions; (2) it is assumed that the anchors and unknown nodes of the sensor network are distributed on the L×L two-dimensional plane and that the nodes are randomly and uniformly distributed; (3) the measurement distance error conforms to a normal distribution and the measurement distance is $d_{ij}^{\prime }$, meaning that we have
(1)$$d_{ij}^{\prime }=d_{ij}\cdot \left(1+\tau \cdot \varepsilon \right),$$
where $d_{ij}$ denotes the real distance from node *i* to node *j*, the distribution of ε is σ(0,1), and τ is the error factor.

The typical DV-Hop approach without range-based positioning [32] includes three steps: calculating the minimum number of hops (MNH) between anchors; calculating the estimated distance from anchors to unknown nodes; and calculating the unknown node coordinate position. The steps of the basic DV-Hop algorithm can be summarized as follows.

In Step 1, the MNH is calculated through the flooding process, in which the anchors broadcast information to each node of the WSN, then the MNH between the unknown nodes and anchor nodes is calculated, as well as the MNH between the anchor nodes themselves, with each communication representing a hop.

In Step 2, the mean distance per hop ${Hop}_{i}$ of each anchor node and the estimated distance $d_{u,i}$ are calculated using Formulas (Equation 2) and (Equation 3), respectively:(2)$${Hop}_{i}=\frac{\sum_{i\neq j}d_{i,j}}{\sum_{i\neq j}h_{i,j}}=\frac{\sum_{i\neq j}\sqrt{{\left(x_{i}-x_{j}\right)}^{2}+{\left(y_{i}-y_{j}\right)}^{2}}}{\sum_{i\neq j}h_{i,j}}$$
where $\left(x_{i},\;y_{i}\right)$ represents the position coordinates of anchor node *i*, $\left(x_{j},\;x_{j}\right)$ represents the position coordinates of anchor node *j*, $h_{i,j}$ denotes the MNH from anchor node *i* to node *j*, and estimated distance $d_{u,i}$ from anchor node *i* to unknown node *u* is formulated as follows:(3)$$d_{u,i}={Hop}_{i}\times h_{u,i}$$
where $h_{u,i}$ represents the MNH from unknown node *u* to anchor node *i*.

In Step 3, the fitness function of the node localization optimization algorithm is calculated using Formula (4):(4)$$f\left(x_{u}\right)=\sum_{i=1}^{m}\sqrt{{\left(x_{u}-x_{i}\right)}^{2}+{\left(y_{u}-y_{i}\right)}^{2}}-d_{u,i}$$
where $\left(x_{u},\;y_{u}\right)$ represents the estimated position coordinates of the *u*-th unknown node, $\left(x_{i},\;y_{i}\right)$ denotes the position coordinates of the *i*-th anchor node, $d_{u,i}$ denotes the estimated distance from unknown node *u* to anchor node *i*, and *m* is the number of anchor nodes.

### 3.2. Node Coverage Optimization (NCO) Problem in WSNs

Assuming that the simulation work area is a two-dimensional space and the side length is *L*, there are *n* detection points that need to be perceived within the sensing radius of the sensor nodes; we use $R_{s}$ and $R_{c}$ to denote the sensing radius and communication radius, respectively. The Euclidean distance d(i, s) between sensor nodes and target points can be calculated as follows:(5)$$d\left(i,\;s\right)=\sqrt{{\left(x_{s}-x_{i}\right)}^{2}+{\left(y_{s}-y_{i}\right)}^{2}}$$
where $\left(x_{i},\;y_{i}\right)$ denotes the position coordinate of the *i*-th target point and $\left(x_{s},\;y_{s}\right)$ denotes the position coordinates of sensor node *s*. The coverage probability *p* using the binary perception model [51,52] from sensor node *s* to target node *i* is
(6)$$p\left(i,\;s\right)=\left\{\begin{array}{c} 0,d\left(i,\;s\right)\geq R_{s},\\ 1,d\left(i,\;s\right)<R_{s}. \end{array}\right.$$

According to the binary perception model, the *x*-axis and *y*-axis of the two-dimensional deployment area can be divided into a step length *q*, that is, each segment length is l=q with an intersection $q^{2}$ of the node deployment area. The deployment node coverage rate of the work area is
(7)$$C\mathrm{ov}=\frac{p_{\mathrm{cov}}}{q^{2}}=\frac{\sum_{i=1}^{S}p(i,\;s)}{q^{2}}.$$

According to the binary perception model and calculation of the node coverage rate of the NCO problem, the mathematical model of this problem can be summarized as a constrained optimization task with four constraints, defined as follows: (8)$$\max f\left(x\right)=Cov,s.t\left\{\begin{array}{c} g_{1}=\sum_{i=1}^{S}p(i,\;s)\geq 0\\ g_{2}=\sum_{i=1}^{S}p\left(i,\;s\right)-q^{2}\geq 0\\ g_{3}=d\left(i,\;s\right)-R_{s}\geq 0\\ g_{4}=S-M\geq 0 \end{array}\right.$$
where p(i, s) denotes the probability of sensing nodes *s* covering and monitoring target nodes *i*, $q^{2}$ is the intersection of the node deployment area, d(i, s) is the Euclidean distance from sensor node *s* to the target monitored node *i*, $R_{s}$ indicates the sensing radius of the sensor node, *M* denotes the number of deployment nodes in the work area by the node-aware range in theory (that is, the sum of the node’s sensing range is equal to the simulation work area), and *S* represents the sensor node number in the monitoring area, which is greater than or equal to the theoretical number of nodes *M*.

## 4. Theory of the Balanced Butterfly Optimizer (BBO)

The BBO is inspired by the influence of both smell and light signals on the foraging process of butterflies. In addition, it is considered that butterflies are photosensitive and that the smell value is generally positive. The specific algorithm modeling processes are population initialization and individual smell and light perception characteristics, which correspond to global and local search, respectively.

### 4.1. Algorithmic Population Initialization

Assuming that the search space is Dim-dimensional, the expression of the initial position of the population is
(9)$$X_{i,j}=\left[\begin{array}{ccc} LB_{1,1} & \cdots & LB_{1,j}\\ \vdots & \ddots & \vdots \\ LB_{i,1} & \cdots & LB_{i,j} \end{array}\right]+\left(\left[\begin{array}{ccc} UB_{1,1} & \cdots & UB_{1,j}\\ \vdots & \ddots & \vdots \\ UB_{i,1} & \cdots & UB_{i,j} \end{array}\right]-\left[\begin{array}{ccc} LB_{1,1} & \cdots & LB_{1,j}\\ \vdots & \ddots & \vdots \\ LB_{i,1} & \cdots & LB_{i,j} \end{array}\right]\right)\cdot \left[\begin{array}{ccc} rand_{1,1} & \cdots & rand_{1,j}\\ \vdots & \ddots & \vdots \\ rand_{i,1} & \cdots & rand_{i,j} \end{array}\right]$$
where $X_{i,j}$ represents the individual initial position, i=1,2,⋯,NP, j=1,2,⋯,Dim, NP denotes the number of initial solutions, Dim denotes the dimension of the problem, $UB_{i,j}$ is the upper boundary value of the search space, $LB_{i,j}$ is the lower boundary value of the search space, and $rand_{i,j}$ represents a random value in (0,1).

### 4.2. Modeling of the Odor- and Light-Sensitive Properties of Butterflies

In nature, there are many kinds of animals with smell perception, butterflies being a typical one; however, different butterfly species have different characteristics. Common butterflies are generally smell-sensing, while light-sensing butterflies mainly include Vanessa Indica [53], Swallowtail [54], and similar varieties; three of these butterfly species shown in Figure 1. Researchers have proven through experiments that these butterflies are affected by both olfactory and visual signals during foraging [55]. Thus, in the design of heuristic algorithms, mathematical modeling based on the conclusions of corresponding biological experiments should be more reasonable and competitive.

(1) Smell-Sensitive Properties

The smell-sensitive properties of butterflies taking into account the exponential diffusion form of odors can be defined as follows:(10)$$Smell_{i}=rand\cdot \left|{\left(F_{i}\right)}^{\alpha }\right|+eps$$
where $Smell_{i}$ represents the smell in the search process, which must be positive in practice and is represented by an absolute value, $F_{i}$ denotes the fitness value obtained by the objective function, rand denotes a random value in (0, 1), α indicates the smell index, which has a value range pf (0.1, 0.6), and eps represents a non-zero infinite decimal, which prevents the smell value from becoming zero during the search process.

(2) Light-Sensitive Properties

The light-sensitive properties of butterflies are affected by the distance between individuals and their food sources or between multiple individuals, which can be defined as follows:(11)$$Light_{i}=Light_{0}\cdot exp(-{D_{ij}}^{2})$$
where $Light_{i}$ denotes the light sensitivity of butterflies and $Light_{0}$ is the initial light sensitivity value, which is set to 1. The light-sensing distance $D_{ij}$ between the searched individual and the food source (or adjacent individuals) can be expressed as
(12)$$D_{ij}=\left|X_{i}^{t}-X_{j}^{t}\right|.$$

### 4.3. Algorithmic Exploitation

The switching probability (sp) is a control parameter between global and local search in the BBO. In nature, there are more butterfly species that rely mainly on scent for foraging or mating than that forage using light. According to this cognitive law, the value of the parameter sp is set to 0.6 in this study. Two search strategies are considered in this study for the proposed algorithm; the best individual is used to choose a search strategy, with the mean position taken into consideration. For sp>rand, meaning that smell-sensitivity is the main search strategy in this stage, the definition is as follows: (13)$$X_{i}^{t+1}=X_{i}^{t}+rand\cdot C\cdot mean\left(G_{best,i}^{t}\right)-X_{i}^{t}\cdot Smell_{i},\;if\;sp>rand$$
where $X_{i}^{t}$ represents the current position of the individuals, rand is a random number in (0, 1), *C* denotes a hyperparameter, which can be set to a random number between (0, 1) or to a constant, $G_{best,i}^{t}$ is the best individual’s search position, and $Smell_{i}$ is the smell sensitivity value in the search process. Alternatively, when light-sensitive is the main search strategy, the position of the individual is updated by
(14)$$X_{i}^{t+1}=X_{i}^{t}+rand\cdot \left(X_{j}^{t}-X_{k}^{t}\right)\cdot Light_{i},\;if\;sp\leq rand$$
where $X_{j}^{t}$ and $X_{k}^{t}$ respectively represent the food source and neighboring individuals of the *t*-th iteration, rand is a random value in (0, 1), and $Light_{i}$ denotes the light sensitivity value in the search process.

### 4.4. Algorithm Computational Complexity Analysis

Different test platforms lead to differences in optimization time consumption for the same optimization method, meaning that the structure of the proposed BBO, that is, the computational complexity, must be analyzed taking this into account. The assumptions are as follows: *N* indicates the population size of the proposed algorithm, *T* represents the maximum number of iterations of the algorithm, and *D* is the dimension for the optimization problem. The computational complexity of the BBO can be summarized as follows: the initialization complexity of the population is O(ND), the fitness computational complexity is O(ND), the sensitivity selection computational complexity is $O(N^{2}logN)$, and the location update of global and local search complexity is $O(N^{2}logN)$. In addition, the complexity of the fitness sorting during the iteration of the algorithm is $O(N^{2})$. Thus, the total computational complexity of BBO is
(15)$$O\left(BBO\right)=O\left(ND\right)+O\left(T\right)\cdot \left(O\left(ND\right)+2\cdot O\left(N^{2}logN\right)+O\left(N^{2}\right)\right).$$

### 4.5. Flowchart and Pseudo-Code of BBO

The flowchart of BBO is presented in Figure 2. There are three main stages of BBO, introduced as follows: Stage 1 represents population initialization and the selection of initial optimal position and fitness values; Stage 2 includes search strategy selection and the process of optimizing the algorithm; and Stage 3 involves selecting the best population of individuals based on the fitness values during the optimization process. The best solution and fitness value are then output after $T_{max}$ iterations.

The main BBO pseudo-code showing the basic operation process is presented in Algorithm 1.
**Algorithm 1:** Pseudo-code of BBO
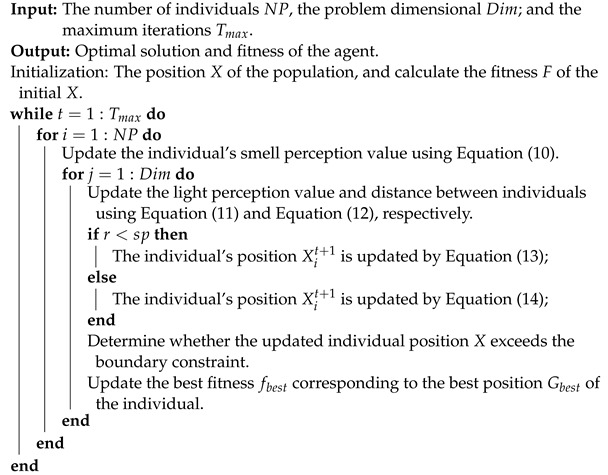


## 5. Analysis of the Numerical Optimization Results

The benchmark functions of the optimization experiment were the CEC function from [17,18], which includes a total of 23 functions, of which F1 to F7 are unimodal functions, F8 to F13 are multimodal functions, and F14 to F23 are fixed-dimensional functions; the corresponding categories are U, M, and Fixed, respectively. Detailed descriptions of all 23 functions are provided in Table 2. The experimental environment was a Windows 10 system with an Intel(R) Core (TM) i5-10210U CPU @2.11 GHz, 16 GB memory, and the Matlab 2018a platform.

### 5.1. Hyperparameter Settings for the Comparison Methods

In this study, 23 benchmark functions were used to verify the performance and effectiveness of the proposed BBO using a random number in (0,1) for the parameter *C*. BBO was compared with the PSO [25], DE [26], GWO [27], ABO [15], BOA [16], HHO [56], and AO algorithms [30]. In addition, the proposed BBO was used for the NLO and NCO problems of WSNs in order to prove its practical applicability. The hyperparameter settings used for the comparison algorithms are presented in Table 3. The code of BBO is released at https://www.researchgate.net/profile/Mengjian-Zhang/research.

Each test benchmark function consisted of 30 independent runs and the $T_{max}$ for the optimization process was set to 1000. The evaluation of the compared SOTA methods included the mean (Mean) and standard deviation (Std) along with the Ranking, as shown in Table 4. Moreover, Table 5 shows the Wilcoxon rank-sum (WSR) test results among the comparison algorithms using a significance level of α=0.05.

### 5.2. Analysis of Benchmark Function Results

The 2D optimization process of the BBO algorithm for certain test functions (TF1, TF4, TF10, and TF23) was described through a visual experiment. The main aim was to qualitatively observe the behavior of the BBO. The shape of the test function, the search history, and the convergence curves of individuals are shown in Figure 3. The search history presents the location history of individual butterflies during an optimized search process. The convergence curve shows the target value of the optimal obtained in each iteration.

The individuals’ search history position in Figure 3 shows that there is a gradual approach to the optimal position during the optimization process. This ensures that the BBO continues to explore and exploit the search space, eventually converging to an optimal point. Compared with the convergence curves for the ABO and BOA in Figure 3, the BBO enhances the initial random population on the test function and ideally improves the accuracy of the approximate optimal value during the iterative process.

Table 4 shows the optimization results of eight comparison algorithms, including a statistical analysis of the Best, Mean, and Ranking of the comparison methods via the statistical results. Here, Best reflects the optimal searchability of BBO for solving numerical optimization problems; the closer the search value is to the value in theory, the better the search performance of the considered approach. The optimization results of the four high-dimensional test functions that reach the theoretical optimal value are TF1, TF3, TF9, and TF11. From the Mean in Table 4, it can be seen that the results of BBO on eight test functions are better than those of the other algorithms, while on nine test functions the optimization results of BBO have the same optimal value as the comparison algorithms. Moreover, the Std objectively shows the stability of the compared methods for solving numerical optimization problems. The results indicate that BBO has high stability and strong generalization ability for numerical optimization problems. From the statistical results of the Friedman test in Table 4, the order of the eight comparison algorithms for the 23 CEC functions is BBO>HHO>AO>DE>GWO>PSO>ABO>BOA.

Notably, the optimization results for the BOA on test functions TF8, TF16, TF19, and TF20 are marked NAN. The reason for this is due to the insufficient design of the BOA. The odor perception value of actual butterflies is positive, and the BOA does not take this into account. It is worth noting that this is a crucial motivation behind our proposed BBO algorithm. In the proposed BBO, the individual’s smell perception value is positive; thus, there is no NAN in the optimization result value.

### 5.3. Analysis of the NLO Problem Results

The BBO algorithm was used to optimize DV-Hop to address the deficiency of the least squares method for the WSN NLO problem. The BBO-DV-Hop localization optimization method obtained better localization accuracy of an unknown node. The anchor nodes were deployed first, then BBO-DV-Hop was utilized to locate the unknown node’s position based on the known coordinates of the anchor nodes. The pseudo-code of the NLO problem via the BBO-DV-Hop algorithm is provided in Algorithm 2.
**Algorithm 2:** Pseudo-code of BBO-DV-Hop
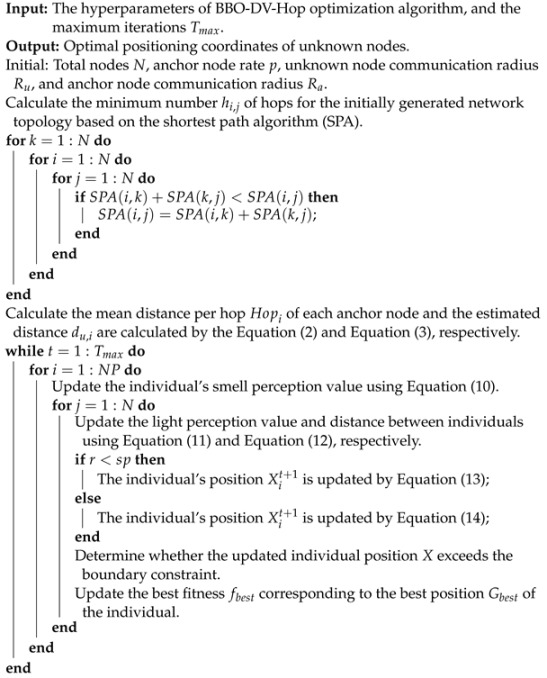


The Average Location Error (ALE) evaluation criteria can be expressed as
(16)$$ALE=\frac{\sqrt{{\left(x_{u}^{estimate}-x_{u}^{reality}\right)}^{2}+{\left(y_{u}^{estimate}-y_{u}^{estimate}\right)}^{2}}}{R\cdot n}$$
where $\left(x_{u}^{estimate},y_{u}^{estimate}\right)$ and $\left(x_{u}^{reality},y_{u}^{reality}\right)$ represent the estimated and actual position coordinates, respectively, of unknown node *u*, *R* denotes the communication radius of the nodes, and *n* represents the number of unknown nodes.

Aiming at the NLO problem in WSNs, the parameters used in this simulation experiment are presented in Table 6. The positions of all nodes are obtained by continuous optimization of the BBO. The deployment area is a 100 m × 100 m square area with 100 nodes, including 20 anchor nodes and 80 unknown nodes, for a rate of anchor nodes to total nodes of 20%. In addition, the communication radius $R_{a}$ of the anchor nodes is set to 30 m and the communication radius $R_{u}$ of the unknown nodes is set to 20 m. Figure 4 shows the simulation results for the initial nodes, with a red “∆” denoting an anchor node and a blue “o” indicating an unknown node. Five comparison algorithms were used to demonstrate the effectiveness of the proposed BBO on the NLO problem in WSNs: PSO, GWO, ABO, BOA, and BBO. Notably, ABO, BOA, and BBO are all inspired by the behaviors of butterflies. The parameters of the comparison methods are shown in Table 3. The maximum iteration $T_{max}$ for solving the NLO problem was set to 200.

From Figure 4, the positions of unknown nodes can be calculated from the known positions of the anchor nodes. The initial anchor node positions directly affect the positioning of the entire node structure, with uniform anchor node positions resulting in a uniform deployment network. In Figure 4a, it can be seen that the network is affected by anchor nodes; local nodes are stacked or located too close together, leading to redundant nodes during the networking process. Although the network nodes in Figure 4b have good connectivity, there are a small number of local nodes. In the upper right corner of Figure 4b only two nodes are connected, leading to the loss of node information. Therefore, in node localization the positions of the initial anchor nodes need to be fully considered; certain movable nodes can be deployed among the unknown nodes for greater robustness, thereby improving the overall life and anti-interference ability of the WSN.

From Figure 5, it can be seen that in terms of the convergence and speed of the compared localization algorithm the order is BBO>ABO>BOA>GWO>PSO. The location error curve of BBO-DV-Hop in the region is relatively smooth after about twenty iterations, which suggests that the proposed BBO-DV-Hop is not likely to fall into local optimal values and has good global convergence ability. On the other hand, the location errors of the PSO and GWO algorithms are relatively high, and gradually become stable after 20 iterations, which indicates that the performance of the proposed BBO-DV-Hop needs to be further modified. The BOA shows large fluctuations before 100 iterations, which suggests that the stability of the BOA needs to be enhanced. In general, while the use of SI algorithms to optimize the NCO problem is a research direction that has received much attention, not all of these algorithms are effective. In actual node positioning deployment, the relocation and the second positioning of mobile nodes could be considered, and design and research could be carried out in combination with optimization algorithms.

In addition, considering the effects on the location of unknown nodes of the number of deployed nodes, rate of anchor nodes, and communication radius, three sets of experiments were designed to compare BBO to PSO, GWO, ABO, and BOA. Figure 6 shows the results of the three experimental parameter settings.

In Figure 6a shows the experimental results for an anchor node rate of 20%, communication radius $R_{a}$ of 30 m, and total number of nodes of 80, 100, 120, 140, 180, and 200. The results show that the trend of LE curves for the compared algorithms is essentially the same. The results obtained with the PSO algorithm fluctuate greatly when the number of nodes is 120 or 140; on the other hand, the BBO algorithm proposed in this study has stable performance, and its LE is the smallest among the compared algorithms.

From Figure 6b, it can be seen that when the number of nodes is 100 and the communication radius $R_{a}$ is 30 m, the anchor point rates are 15%, 20%, 25%, 30%, 35%, 40%, and 45%, respectively, in the node localization optimization simulation experiment. The results show that the LE curves of the compared algorithms are essentially the same. The LE decreases with an increasing anchor node rate, demonstrating that a higher number of anchor nodes allows for more accurate positioning of the unknown nodes. Among the compared algorithms, the location error of PSO is essentially unchanged for anchor node rates of 20%, 25%, and 30%. The proposed BBO algorithm has stable performance, and its location error is the smallest among the compared algorithms.

Figure 6c shows the results of the simulated node localization optimization experiments when the number of nodes is 100, the anchor node rate is 20%, and the communication radius $R_{a}$ is 20 m, 25 m, 30 m, 35 m, 40 m, 45 m, and 50 m. These results show that the LE curves of the compared algorithms are essentially the same. The location error decreases with the increase in the anchor node communication radius when $R_{a}$ is less than or equal to 30 m. Moreover, the location error shows an overall upward trend when $R_{a}$ exceeds 30 m. Again, the proposed BBO algorithm has stable performance and its LE is the smallest among the comparison algorithms.

### 5.4. Analysis of the NCO Problem Results

To verify the effectiveness of the BBO with the parameter C=1 for solving the NCO problem, we used a deployment area of 100 m × 100 m in the simulation experiments and we performed the following experiments: (1) by randomly deploying 40 and 45 nodes with sensing radius $R_{s}=10$ m and communication radius $R_{c}=20$ m, we analyzed the coverage optimization performance of BBO over different numbers of iterations using both the time and coverage; (2) BBO’s performance when deploying a different number of nodes was analyzed using random deployment of nodes with sensing radius $R_{s}=10$ m and communication radius $R_{c}=20$ m over 100 iterations; (3) through random deployment, BBO’s performance with a different communication radius was analyzed with 20 nodes over 100 iterations. The pseudo-code of the NCO problem based on the BBO algorithm can be seen in Algorithm 3.
**Algorithm 3:** Pseudo-code of BBO for the NCO problem
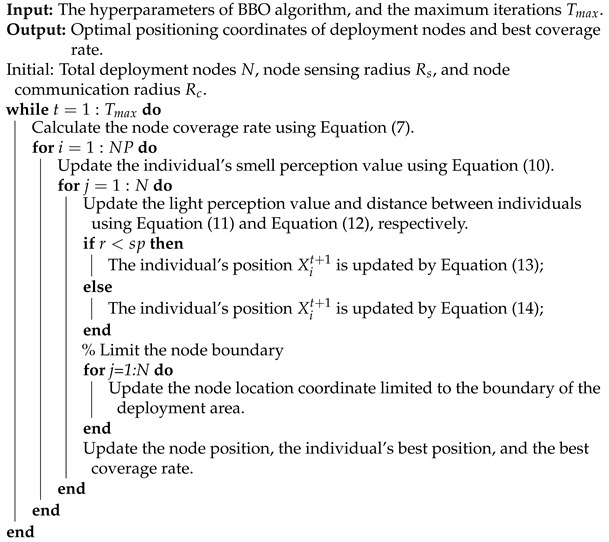


From Table 7, when the number of nodes is 40 and the sensing radius is 10 m, the coverage results based on the BBO algorithm corresponding to 100, 150, 200, and 500 iterations are 95.79%, 96.62%, 96.67%, and 96.99%, respectively. The time consumption of the best position of the output node is 17.17 s, 24.48 s, 31.50 s, and 79.56 s, respectively. The coverage and time consumption of nodes show an increasing trend. The coverage of the points increases by 0.82%, 0.05%, and 0.32%, respectively, year-on-year, while the year-on-year time consumption increases by 7.32 s, 7.01 s, and 48.06 s, respectively. When the number of nodes is 45 and the sensing radius is 10 m, the coverage optimization results based on the BBO algorithm corresponding to 100, 150, 200, and 500 iterations are 97.76%, 98.50%, 98.56%, and 98.82%, respectively. The time consumption of the best position of the output node is 21.87 s, 28.38 s, 37.29 s, and 86.35 s, respectively. The coverage and time consumption of nodes show an increasing trend. The year-on-year coverage of the points increases by 0.75%, 0.06%, and 0.26%, respectively, while the year-on-year time consumption increases by 6.51 s, 8.91 s, and 49.06 s, respectively. Based on the two scenarios with the same number of iterations, when the number of nodes increases, the node coverage optimization of BBO is significantly improved, although the time consumption is increased.

From Table 8, it can be seen that when the sensing radius is 10 m and the deployment area is 100 m × 100 m, the coverage optimization results based on the BBO algorithm correspond to a node coverage of 95.79%, 97.76%, and 98.29% for 40, 45, and 50 nodes, respectively. The time consumption of the best node position is 17.17 s, 21.87 s, and 25.83 s, respectively. The coverage is 1.96 percentage points higher with 45 nodes than with 40 nodes, and is 0.54 percentage points higher with 50 nodes than with 45 nodes. The year-on-year time consumption increased by 4.70 s and 3.96 s, respectively. In addition, these results show that the optimized coverage is significantly improved when the number of deployment nodes is increased within a certain range. The coverage growth is slow until saturation when the number of nodes is increased in transition.

Table 9 shows that when the deployment area is 100 m × 100 m with 20 deployment nodes, the coverage rates obtained with BBO corresponding to a sensing radius of 13 m, 14 m, and 15 m are 91.67%, 97.15%, and 99.38%, respectively. The time consumption of the best position of the output node is 6.07 s, 6.13 s, and 5.67 s, respectively. The year-on-year node coverage of the points increases by 6.48% and 2.23%, respectively. The time consumption of the BBO algorithm for node position optimization is less sensitive to the change in the sensing radius, especially when the perception radius is 15 m, and the simulation time consumption decreases by 0.46 s year-on-year. In addition, these experimental results show that when the number of nodes is fixed, an increase in the sensor radius increases coverage and decreases time consumption. Thus, when the deployment area is the same and the sensing radius is excessively large, the number of deployment nodes should be reduced.

To assess the superiority of the proposed BBO algorithm for solving the NCO problem, we selected four SI algorithms as comparison methods namely, PSO, GWO, ABO, and BOA, as well as DSA [29] and two improved butterfly optimization algorithms, HPSBA [51] and FBA [19]. The simulation parameters for the node coverage optimization problem were as follows: the deployment area was 100 m × 100 m, the number of deployment nodes was 20, and the number of iterations was set to 100. The results on the NCO problem with a changing sensing radius (13 m, 14 m, 15 m) are shown in Table 10.

Table 10 shows that for a sensing radius of 14 m, the results with the proposed BBO are significantly better than with the compared methods. The coverage rate of BBO is 91.67%; when the sensing radius is 13 m or 15 m, the coverage rate for the NCO problem achieved with BBO is higher than that achieved with the other algorithms. The performance of BBO in solving the NCO problems is significantly modified, which indicates that it has high application significance. Compared with the BOA, the node coverage rate of BBO with a sensing radius $R_{s}$ of 13 m, 14 m, and 15 m increased by 13.40 percentage points, 12.11 percentage points, and 8.18 percentage points, respectively. In addition, Figure 7 and Figure 8 show the coverage curves for node coverage optimization achieved by the compared algorithms (PSO, GWO, ABO, BOA, DSA, HPSBA, FBA, and BBO).

## 6. Conclusions

In this paper we propose a novel algorithm, BBO, inspired by the fact that butterflies have both smell-sensitive and light-sensitive characteristics. We respectively translate these smell and light-sensitive characteristics into the local and global search strategies of the proposed algorithm. In addition, the value of an individual’s smell-sensitivity is positive, which is a point that cannot be neglected. Our results show that the proposed BBO has better performance in terms of global search capability and stability than other algorithms used for comparison. In addition, our simulation results indicate that the BBO-DV-Hop localization optimization algorithm proposed in this study has good stability and accuracy on the NLO problem for WSNs. The proposed BBO has superior performance on the NCO problem as well. In both cases, the position of the initial anchor nodes needs to be fully considered in order to ensure the robustness of the network after positioning. In future studies, certain movable nodes can be deployed as unknown nodes to improve the overall lifetime and anti-interference ability of the WSN.

The performance of the proposed BBO can be enhanced to solve high-dimensional optimization problems. In-depth study could lead to more effective improvement strategies which can be applied to engineering optimization, IoT, feature selection, and more. Among these applications, node deployment, routing, dynamic networking optimization, and other problems involving WSNs [57,58] can be solved by SI optimization algorithms. Currently, research on 3D WSNs is a hot topic, with contexts including 3D space, underwater scenarios, complex mountains, forest fire monitoring, and more.

## Figures and Tables

**Figure 1 biomimetics-08-00393-f001:**
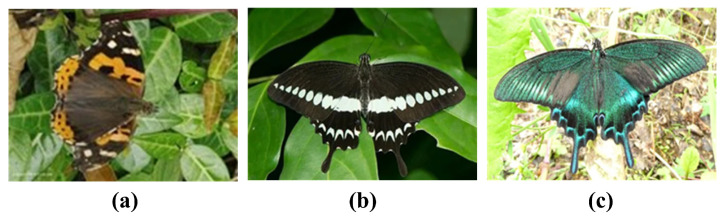
Butterflies that use both olfactory and visual signals for foraging: (**a**) Vanessa Indica; (**b**) Swallowtail; (**c**) Papilio Maackii.

**Figure 2 biomimetics-08-00393-f002:**
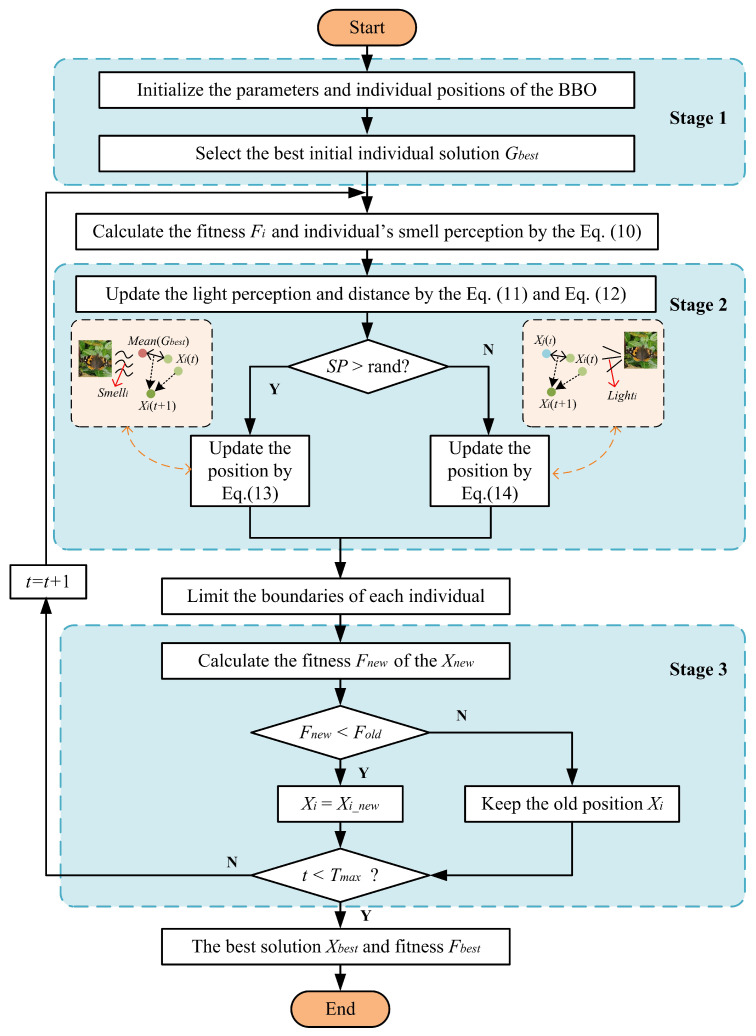
The flowchart of the proposed BBO.

**Figure 3 biomimetics-08-00393-f003:**
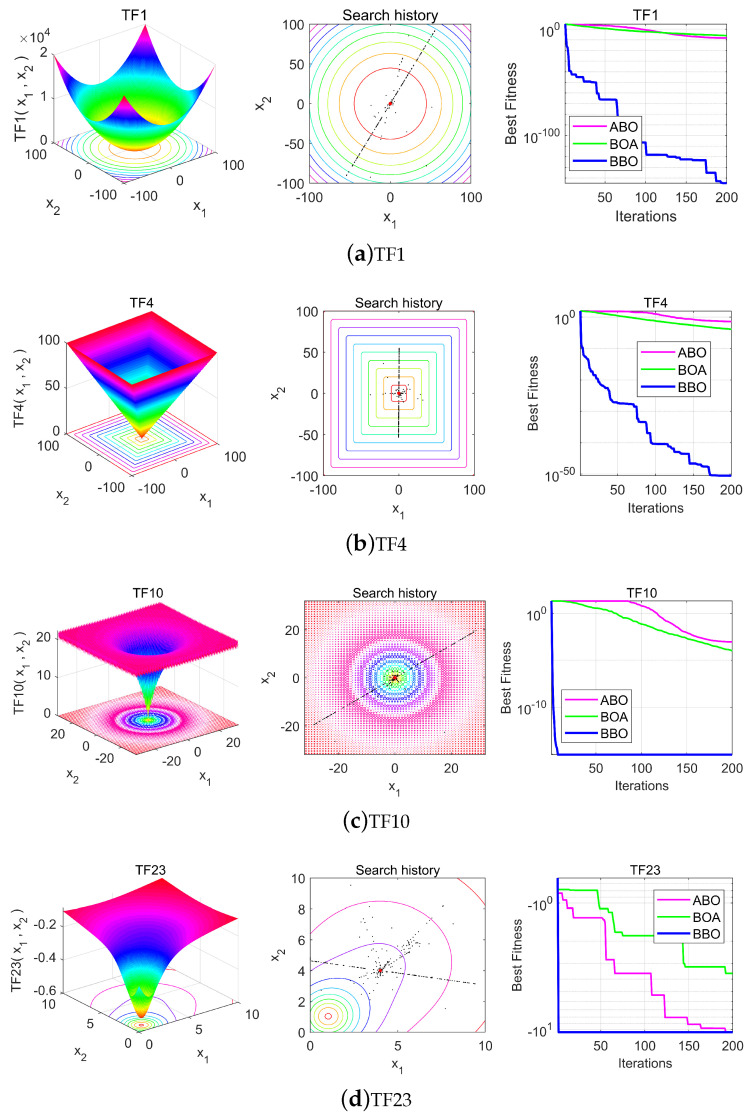
Individual historical positions with two-dimensional coordinate and convergence curves of the search process.

**Figure 4 biomimetics-08-00393-f004:**
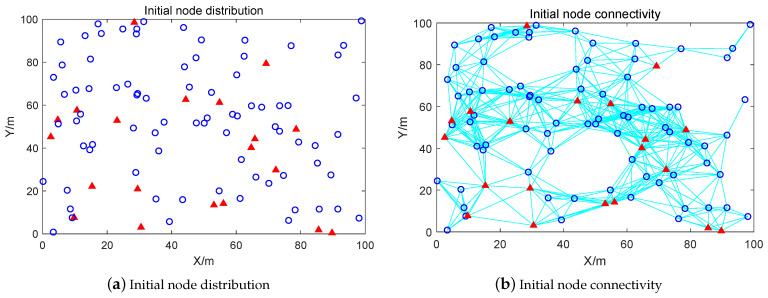
Schematic of initial node localization optimization and connectivity.

**Figure 5 biomimetics-08-00393-f005:**
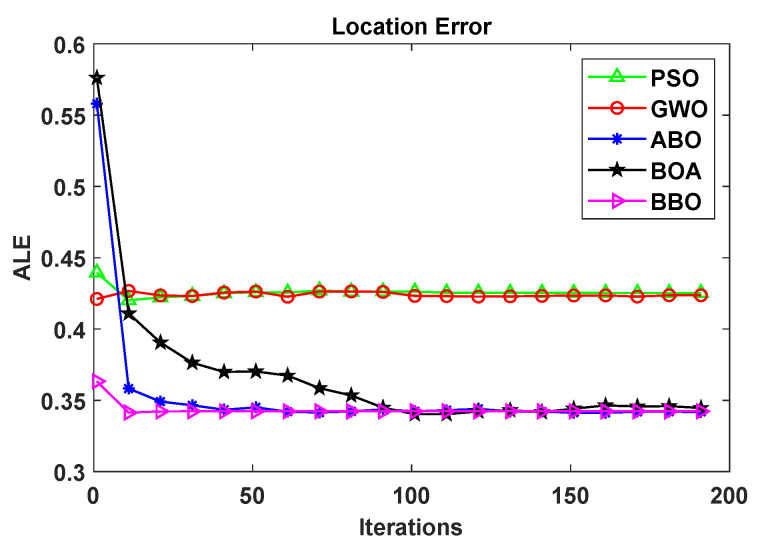
Comparison curves of the location error of the compared SI algorithms over 200 iterations.

**Figure 6 biomimetics-08-00393-f006:**
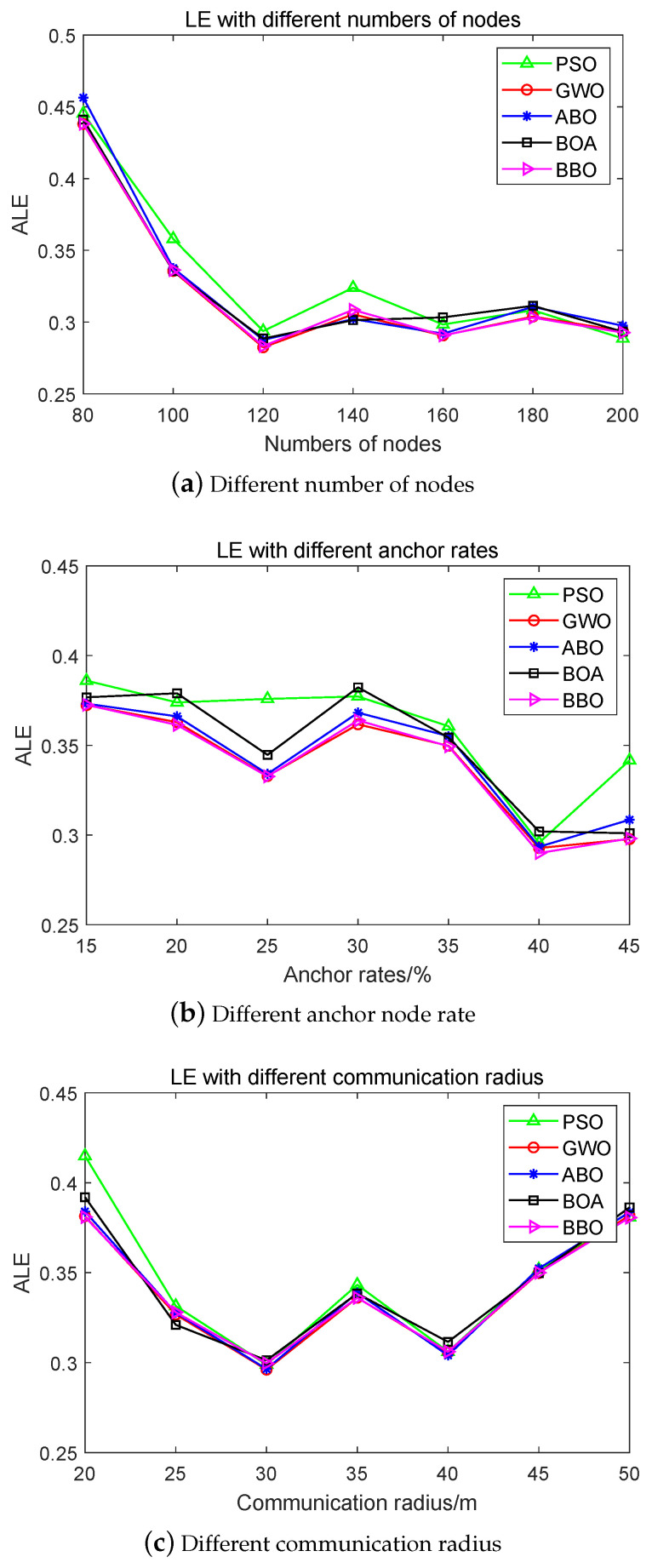
Results of location errors (LE) with different experimental parameters: (**a**) different number of nodes; (**b**) different anchor node rate; and (**c**) different communication radius.

**Figure 7 biomimetics-08-00393-f007:**
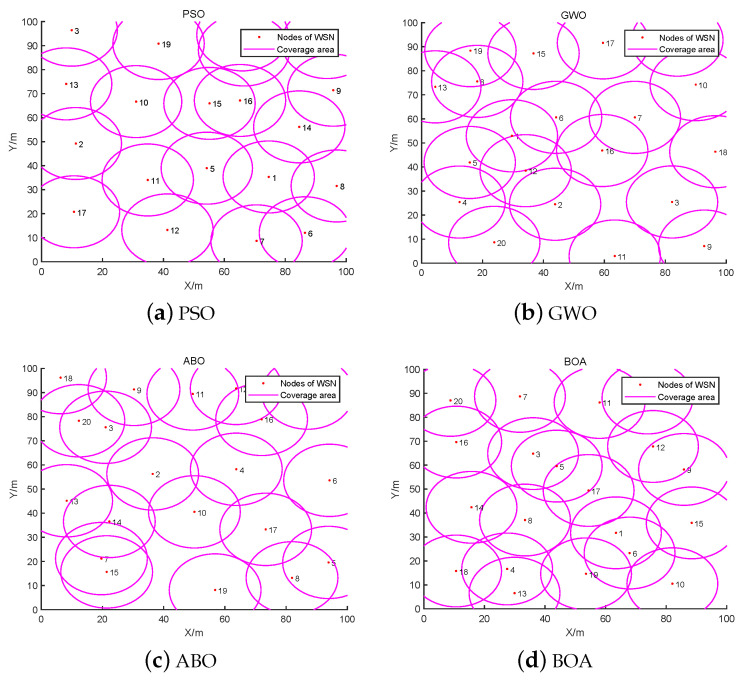
Schematic of node deployment optimization with different SI algorithms.

**Figure 8 biomimetics-08-00393-f008:**
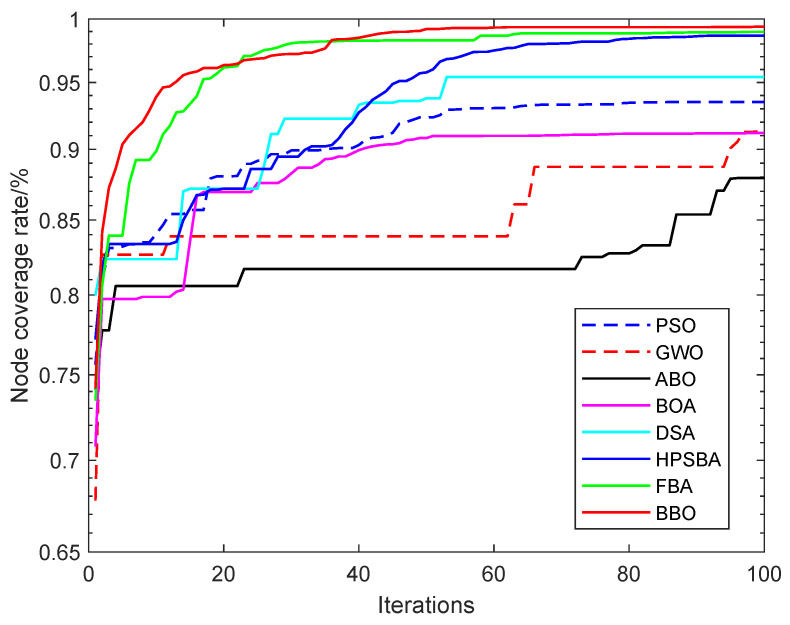
Node coverage rate curves of the compared SI algorithms over 100 iterations.

**Table 1 biomimetics-08-00393-t001:** A brief summary of NLO and NCO problems using SI algorithms.

Author	Problem	Algorithm	Title
Shi et al. [43]	NLO	Particle swarm optimization (PSO)	An improved DV-Hop scheme based on path matching and particle swarm optimization algorithm
Han et al. [44]	NLO	DE algorithm	Enhancing the sensor node localization algorithm based on improved DV-hop and DE algorithms in wireless sensor networks
Zhang et al. [45]	NLO	Enhanced sparrow search algorithm	Multi-strategy improved sparrow search algorithm for solving the node localization problem in heterogeneous wireless sensor networks
Ghafour et al. [46]	NLO	Enhanced squirrel search algorithm	Improved DV-Hop based on squirrel search algorithm for localization in wireless sensor networks
Yang et al. [47]	NCO	Improved Firefly algorithm (FA)	Deploying charging nodes in wireless rechargeable sensor networks based on improved firefly algorithm
Wang et al. [48]	NCO	Resampled PSO algorithm	Coverage control of sensor networks in IoT based on RPSO
Miao et al. [49]	NCO	GWO-EH algorithm	Grey wolf optimizer with an enhanced hierarchy and its application to the wireless sensor network coverage optimization problem
Wang et al. [50]	NCO	Wolf pack algorithm (WPA)	A novel topology-based coverage-oriented strategy optimization of wireless sensor networks
Zhang et al. [51]	NCO	hybrid HPSBA	HPSBA, a modified hybrid framework with convergence analysis for solving the wireless sensor network coverage optimization problem

**Table 2 biomimetics-08-00393-t002:** Descriptions of the 23 benchmark functions.

Formula	SR	Dim	$f_{\min}$	Category
${TF}_{1}=\sum_{i=1}^{Dim}x_{i}^{2}$	[−100, 100]	30	0	U
${TF}_{2}=\sum_{i=1}^{Dim}\left|x_{i}\right|+\prod_{i=1}^{Dim}\left|x_{i}\right|$	[−10, 10]	30	0	U
${TF}_{3}=\sum_{i=1}^{Dim}{\left(\sum_{j=1}^{i}x_{j}\right)}^{2}$	[-100, 100]	30	0	U
${TF}_{4}=\max\left\{\left|x_{i}\right|,1\leq i\leq Dim\right\}$	[−100, 100]	30	0	U
${TF}_{5}=\sum_{i=1}^{Dim}\left(100{\left(x_{i+1}-x_{i}^{2}\right)}^{2}+{\left(x_{i}-1\right)}^{2}\right)$	[−30,30]	30	0	U
${TF}_{6}=\sum_{i=1}^{Dim}{\left(x_{i}+0.5\right)}^{2}$	[-100, 100]	30	0	U
${TF}_{7}=\sum_{i=1}^{Dim}ix_{i}^{4}+rand\left(0,1\right)$	[−1.28, 1.28]	30	0	U
${TF}_{8}\left(x\right)=\sum_{i=1}^{n}-x_{i}\sin(\sqrt{\left|x_{i}\right|})$	[−500, 500]	30	−12,569.487	M
${TF}_{9}=\sum_{i=1}^{Dim}\left(x_{i}^{2}-10\cos\left(2\pi x_{i}\right)+10\right)$	[−5.12, 5.12]	30	0	M
${TF}_{10}=-20\exp\left(-0.2\sqrt{\frac{1}{Dim}\sum_{i=1}^{Dim}x_{i}^{2}}\right)-\exp\left(\frac{1}{Dim}\sum_{i=1}^{Dim}\cos(2\pi x_{i})\right)+20+e$	[−32, 32]	30	0	M
${TF}_{11}=\frac{1}{4000}\sum_{i=1}^{Dim}x_{i}^{2}-\prod_{i=1}^{Dim}\cos\left(\frac{x_{i}}{\sqrt{i}}\right)+1$	[−600, 600]	30	0	M
${TF}_{12}=\frac{\pi }{Dim}\left\{\sum_{i=1}^{Dim-1}{\left(y_{i}-1\right)}^{2}\left[1+10\sin^{2}\left(\pi y_{i+1}\right)\right]+{\left(y_{Dim-1}\right)}^{2}+10\sin^{2}\left(\pi y_{1}\right)\right\}+\sum_{i=1}^{Dim}u(x_{i},10,100,4),y_{i}=1+\frac{x_{i}+1}{4},u_{y_{i},a,k,m}=\left\{\begin{array}{c} k{\left(x_{i}-a\right)}^{m},x_{i}>a,\\ 0,-a\leq x_{i}\leq a,\\ k{\left(-x_{i}-a\right)}^{m},x_{i}<a \end{array}\right.$	[−50, 50]	30	0	M
${TF}_{13}=\frac{1}{10}\left\{\sin^{2}\left(\pi x_{1}\right)+\sum_{i=1}^{Dim-1}{\left(x_{i}-1\right)}^{2}\left[1+\sin^{2}\left(3\pi x_{i+1}\right)\right]+{\left(x_{Dim-1}\right)}^{2}\left(1+\sin^{2}\left(2\pi x_{i+1}\right)\right)\right\}+\sum_{i=1}^{Dim}u(x_{i},5,100,4)$	[−5, 5]	30	0	M
${TF}_{14}\left(x\right)={\left(\frac{1}{500}+\sum_{j=1}^{25}\frac{1}{j+\sum_{i=1}^{2}{\left(x_{i}-a_{ij}\right)}^{6}}\right)}^{-1}$	[−65, 65]	2	1	Fixed
${TF}_{15}\left(x\right)=\sum_{i=1}^{11}{\left[a_{i}-\frac{x_{1}\left(b_{i}^{2}+b_{i}x_{2}\right)}{b_{i}^{2}+b_{i}x_{3}+x_{4}}\right]}^{2}$	[−5, 5]	4	0.00030	Fixed
${TF}_{16}\left(x\right)=4x_{1}^{2}-2.1x_{1}^{4}+\frac{1}{3}x_{1}^{6}+x_{1}x_{2}-4x_{2}^{2}+4x_{2}^{4}$	[−5, 5]	2	−1.0316	Fixed
${TF}_{17}\left(x\right)={\left(x_{2}-\frac{5.1}{4\pi^{2}}x_{1}^{2}+\frac{5}{\pi }x_{1}-6\right)}^{2}+10\left(1-\frac{1}{8\pi }\right)\cos x_{1}+10$	[−5, 5]	2	0.3980	Fixed
${TF}_{18}\left(x\right)=\left[1+{\left(x_{1}+x_{2}+1\right)}^{2}\left(19-14x_{1}+3x_{1}^{2}-14x_{2}+6x_{1}x_{2}+3x_{2}^{2}\right)\right]\times \left[30+{\left(2x_{1}-3x_{2}\right)}^{2}\times \left(18-32x_{1}+12x_{1}^{2}+48x_{2}-36x_{1}x_{2}+27x_{2}^{2}\right)\right]$	[−2, 2]	2	3	Fixed
${TF}_{19}\left(x\right)=-\sum_{i=1}^{4}c_{i}\exp\left(-\sum_{j=1}^{3}a_{ij}{\left(x_{j}-p_{ij}\right)}^{2}\right)$	[1, 3]	3	−3.86	Fixed
${TF}_{20}\left(x\right)=-\sum_{i=1}^{4}c_{i}\exp\left(-\sum_{j=1}^{6}a_{ij}{\left(x_{j}-p_{ij}\right)}^{2}\right)$	[0, 1]	5	−3.32	Fixed
${TF}_{21}\left(x\right)=-\sum_{i=1}^{5}{\left[\left(X-a_{i}\right)\left(X-a_{i}\right)+c_{i}\right]}^{-1}$	[0, 10]	4	−10.1532	Fixed
${TF}_{22}\left(x\right)=-\sum_{i=1}^{7}{\left[\left(X-a_{i}\right)\left(X-a_{i}\right)+c_{i}\right]}^{-1}$	[0, 10]	4	−10.4028	Fixed
${TF}_{23}\left(x\right)=-\sum_{i=1}^{10}{\left[\left(X-a_{i}\right)\left(X-a_{i}\right)+c_{i}\right]}^{-1}$	[0, 10]	4	−10.5363	Fixed

**Table 3 biomimetics-08-00393-t003:** Hyperparameter settings used for the comparison methods.

Methods	Value
PSO	$NP=30,c_{1}=c_{2}=2,\omega_{min}=0.2,\omega_{max}=0.9$
DE	NP=30,F=0.85,CR=0.7
GWO	$NP=30,a_{first}=2,a_{final}=0$
ABO	$NP=30,a_{first}=2,a_{final}=0$
BOA	$NP=30,a=0.1,c\left(0\right)=0.01,p=0.5$
HHO	NP=30,β=1.5,E∈(0,2)
AO	NP=30,α=0.1,δ=0.1
BBO	$NP=30,\alpha =0.1,{Light}_{0}=1,sp=0.6$

**Table 4 biomimetics-08-00393-t004:** Means and standard deviations of the compared SI methods for all 23 functions.

Function	Item	PSO	DE	GWO	ABO	BOA	HHO	AO	BBO
TF1	Mean	2.14E-03	6.13E-07	2.79E-59	7.25E-56	4.98E-13	2.91E-184	4.49E-199	0.00E+00
	Std	2.29E-03	4.05E-07	5.32E-59	3.12E-55	5.14E-14	0.00E+00	0.00E+00	0.00E+00
TF2	Mean	3.74E-03	1.94E-04	1.19E-34	1.54E-37	2.31E-10	1.22E-94	1.77E-118	2.35E-129
	Std	4.23E-03	9.01E-05	1.12E-34	2.88E-37	6.15E-11	6.45E-94	9.68E-118	1.06E-128
TF3	Mean	4.11E+03	1.07E+04	2.71E-15	3.00E-20	4.27E-13	6.86E-153	1.28E-199	0.00E+00
	Std	1.26E+03	4.10E+03	9.11E-15	1.30E-19	3.78E-14	3.69E-152	0.00E+00	0.00E+00
TF4	Mean	1.39E+01	3.78E+00	1.72E-14	7.53E-14	3.24E-10	6.38E-94	1.92E-98	1.77E-74
	Std	2.97E+00	2.09E+00	3.03E-14	2.03E-13	2.61E-11	3.17E-93	1.05E-97	9.71E-74
TF5	Mean	1.48E+02	5.05E+01	2.68E+01	2.73E+01	2.89E+01	3.55E-03	1.08E-03	2.85E+01
	Std	1.27E+02	3.81E+01	7.05E-01	1.00E+00	2.92E-02	7.62E-03	1.82E-03	2.54E-01
TF6	Mean	2.61E-03	6.94E-07	6.03E-01	2.22E+00	4.83E+00	2.19E-05	3.15E-05	1.08E-11
	Std	3.21E-03	4.40E-07	3.25E-01	3.86E-01	5.64E-01	3.03E-05	6.35E-05	1.60E-11
TF7	Mean	7.99E-02	3.33E-02	8.76E-04	7.77E-04	1.26E-03	5.63E-05	4.80E-05	7.59E-06
	Std	2.98E-02	1.10E-02	5.51E-04	4.40E-04	5.76E-04	4.03E-05	4.47E-05	5.10E-06
TF8	Mean	−1.12E+04	−9.26E+03	−6.12E+03	−6.18E+03	NAN	−1.26E+04	−9.48E+03	−7.85E+03
	Std	3.08E+02	1.39E+03	5.98E+02	5.84E+02	NAN	2.83E-01	3.67E+03	4.16E+03
TF9	Mean	6.19E+01	1.36E+02	5.30E-01	0.00E+00	7.13E+01	0.00E+00	0.00E+00	0.00E+00
	Std	1.22E+01	2.26E+01	2.54E+00	0.00E+00	9.51E+01	0.00E+00	0.00E+00	0.00E+00
TF10	Mean	2.77E-01	2.63E+00	1.68E-14	2.03E+00	1.22E-10	8.88E-16	8.88E-16	8.88E-16
	Std	5.36E-01	6.81E+00	2.59E-15	6.21E+00	6.53E-11	0.00E+00	0.00E+00	0.00E+00
TF11	Mean	2.65E-02	2.67E-03	2.19E-03	1.32E-03	1.87E-15	0.00E+00	0.00E+00	0.00E+00
	Std	2.88E-02	5.97E-03	5.93E-03	5.19E-03	3.63E-15	0.00E+00	0.00E+00	0.00E+00
TF12	Mean	8.68E-01	2.07E-02	4.00E-02	1.60E-01	3.64E-01	1.81E-06	6.40E-07	1.60E-12
	Std	1.01E+00	7.89E-02	2.37E-02	1.02E-01	1.22E-01	2.50E-06	1.07E-06	2.92E-12
TF13	Mean	6.70E-01	4.24E-04	4.77E-01	1.56E+00	2.45E+00	1.60E-05	1.54E-05	2.63E-01
	Std	1.41E+00	2.02E-03	2.24E-01	2.41E-01	4.69E-01	2.23E-05	3.05E-05	8.01E-01
TF14	Mean	0.998004	1.130541	4.418589	3.541092	1.068958	1.097407	2.374092	1.129753
	Std	4.12E-17	3.44E-01	4.58E+00	4.11E+00	2.56E-01	3.03E-01	2.71E+00	7.22E-01
TF15	Mean	0.003171	0.004441	0.004384	0.000535	0.000356	0.000362	0.000448	0.000307
	Std	6.97E-03	8.10E-03	8.13E-03	3.89E-04	4.33E-05	1.64E-04	8.76E-05	1.83E-19
TF16	Mean	−1.031628	−1.031628	−1.031628	−1.031628	NAN	−1.031628	−1.031495	−1.031628
	Std	6.45E-16	6.78E-16	7.86E-09	5.70E-09	NAN	3.33E-11	1.15E-04	6.78E-16
TF17	Mean	0.397887	0.397887	0.477671	0.397888	0.398020	0.397888	0.397996	0.397887
	Std	0.00E+00	0.00E+00	4.37E-01	3.78E-07	1.82E-04	7.21E-07	1.39E-04	0.00E+00
TF18	Mean	3	3	3.000009	3.000000	3.005992	3	3.011503	3
	Std	1.40E-15	2.12E-15	1.19E-05	3.20E-07	9.81E-03	2.44E-08	1.44E-02	1.90E-15
TF19	Mean	−3.862782	−3.862782	−3.861725	−3.857002	NAN	−3.861322	−3.858885	−3.862782
	Std	2.63E-15	2.71E-15	2.28E-03	3.54E-03	NAN	1.98E-03	2.47E-03	2.71E-15
TF20	Mean	−3.282364	−3.266512	−3.262065	−3.095172	NAN	−3.121452	−3.198913	−3.321995
	Std	5.70E-02	6.03E-02	6.83E-02	1.98E-01	NAN	1.12E-01	8.21E-02	1.34E-15
TF21	Mean	−7.040316	−7.805056	−9.814474	−7.363414	−5.790008	−5.222809	−10.144644	−6.908833
	Std	2.86E+00	3.02E+00	1.29E+00	3.47E+00	1.07E+00	9.22E-01	1.67E-02	3.59E+00
TF22	Mean	−8.307454	−9.220698	−10.402433	−8.667653	−5.925582	−5.086337	−10.400402	−9.031198
	Std	2.86E+00	2.44E+00	2.73E-04	3.33E+00	1.34E+00	1.96E-03	3.61E-03	2.84E+00
TF23	Mean	−10.17745	−10.53641	−10.53592	−9.09307	−5.88811	−5.03705	−10.53198	−9.33568
	Std	1.37E+00	2.21E-15	2.92E-04	2.73E+00	1.27E+00	4.94E-01	7.52E-03	2.76E+00
Mean/+/−/=		1/18/4	1/18/4	1/21/1	0/21/2	0/23/0	1/17/5	3/18/2	8/9/6
Fridman Rank		5.86	5.04	5.57	5.87	7.58	4.66	4.76	2.59
Total Rank		6	4	5	7	8	2	3	1

**Table 5 biomimetics-08-00393-t005:** Results of WSR test for the compared SI methods.

Function	BBO vs. PSO	BBO vs. DE	BBO vs. GWO	BBO vs. ABO	BBO vs. BOA	BBO vs. HHO	BBO vs. AO
TF1	1.73E-06(1)	1.73E-06(1)	1.73E-06(1)	1.73E-06(1)	1.73E-06(1)	1.73E-06(1)	1.73E-06(1)
TF2	1.73E-06(1)	1.73E-06(1)	1.73E-06(1)	1.73E-06(1)	1.73E-06(1)	1.73E-06(1)	1.02E-01(0)
TF3	1.73E-06(1)	1.73E-06(1)	1.73E-06(1)	1.73E-06(1)	1.73E-06(1)	1.73E-06(1)	1.73E-06(1)
TF4	1.73E-06(1)	1.73E-06(1)	1.73E-06(1)	1.73E-06(1)	1.73E-06(1)	3.11E-05(1)	1.48E-04(1)
TF5	2.13E-06(1)	3.60E-01(0)	1.92E-06(1)	2.16E-05(1)	1.73E-06(1)	1.73E-06(1)	1.73E-06(1)
TF6	1.73E-06(1)	1.73E-06(1)	1.73E-06(1)	1.73E-06(1)	1.73E-06(1)	1.73E-06(1)	1.73E-06(1)
TF7	1.73E-06(1)	1.73E-06(1)	1.73E-06(1)	1.73E-06(1)	1.73E-06(1)	2.88E-06(1)	7.69E-06(1)
TF8	1.48E-03(1)	8.59E-02(0)	7.52E-02(0)	6.87E-02(0)	NAN(1)	1.48E-03(1)	1.92E-01(0)
TF9	1.73E-06(1)	1.73E-06(1)	1.56E-02(1)	1(0)	6.10E-05(1)	1(0)	1(0)
TF10	1.73E-06(1)	1.73E-06(1)	6.83E-07(1)	4.15E-07(1)	1.73E-06(1)	1(0)	1(0)
TF11	1.73E-06(1)	1.73E-06(1)	1.25E-01(0)	5.00E-01(0)	8.77E-05(1)	1(0)	1(0)
TF12	1.73E-06(1)	1.73E-06(1)	1.73E-06(1)	1.73E-06(1)	1.73E-06(1)	1.73E-06(1)	1.73E-06(1)
TF13	1.48E-03(1)	2.77E-03(1)	2.77E-03(1)	6.98E-06(1)	3.18E-06(1)	2.77E-03(1)	2.77E-03(1)
TF14	1(0)	6.45E-04(1)	7.69E-06(1)	1.24E-05(1)	3.11E-05(1)	3.11E-05(1)	1.73E-06(1)
TF15	1.73E-06(1)	1.54E-04(1)	1.73E-06(1)	1.73E-06(1)	1.73E-06(1)	1.73E-06(1)	1.73E-06(1)
TF16	1(0)	1(0)	1.73E-06(1)	1.73E-06(1)	NAN(1)	2.56E-06(1)	1.73E-06(1)
TF17	1(0)	1(0)	1.73E-06(1)	1.73E-06(1)	1.73E-06(1)	8.30E-06(1)	1.73E-06(1)
TF18	5.86E-03(1)	5.08E-01(0)	1.73E-06(1)	1.73E-06(1)	1.73E-06(1)	1.73E-06(1)	1.73E-06(1)
TF19	1(0)	1(0)	1.73E-06(1)	1.73E-06(1)	NAN(1)	1.73E-06(1)	1.73E-06(1)
TF20	1.95E-03(1)	6.10E-05(1)	1.73E-06(1)	1.73E-06(1)	NAN(1)	1.73E-06(1)	1.73E-06(1)
TF21	9.86E-01(0)	3.65E-01(0)	6.27E-02(0)	6.88E-01(0)	5.71E-02(0)	7.27E-03(1)	4.72E-02(0)
TF22	6.03E-01(0)	8.24E-01(0)	1.65E-01(0)	4.49E-02(0)	1.36E-04(1)	1.02E-05(1)	1.65E-01(0)
TF23	1.09E-01(0)	6.25E-02(0)	5.71E-02(0)	7.73E-03(1)	1.64E-05(1)	6.98E-06(1)	5.71E-02(0)
H(0/1)	7/16	9/14	5/18	5/18	1/22	3/20	8/15

**Table 6 biomimetics-08-00393-t006:** Simulation parameter settings for the NLO problem.

Parameters	Value
Deployment area/m	100 × 100
The communication radius $R_{a}$ of the anchor nodes/m	30
The communication radius $R_{u}$ of unknown nodes/m	20
Anchor node rate/%	20
The NP of individuals	30
Total number of nodes *N*	100

**Table 7 biomimetics-08-00393-t007:** Coverage with different numbers of iterations.

Item	*N* = 40, $R_{s}$ = 10 m				*N* = 45, $R_{s}$ = 10 m			
Iterations	100	150	200	500	100	150	200	500
Cov/%	95.79	96.62	96.67	96.99	97.76	98.50	98.56	98.82
Time/s	17.17	24.48	31.50	79.56	21.87	28.38	37.29	86.35
Percentage point increase in coverage/%	/	0.82	0.05	0.32	/	0.75	0.06	0.26
Increase in time/s	/	7.32	7.01	48.06	/	6.51	8.91	49.06

**Table 8 biomimetics-08-00393-t008:** Node coverage rate with different number of deployment nodes.

Node Number	40	45	50
Cov/%	95.79	97.76 (+1.96)	98.29 (+0.54)
Time/s	17.17	21.87 (+4.70)	25.83 (+3.96)

**Table 9 biomimetics-08-00393-t009:** Node coverage rates with changes in the sensing radius.

$R_{s}$	13	14	15
Cov /%	91.67	97.15 (+6.48)	99.38 (+2.23)
Time /s	6.07	6.13 (+0.06)	5.67 (−0.46)

**Table 10 biomimetics-08-00393-t010:** Comparison of algorithms on the NCO problem with a changing sensing radius.

Method	13 m	14 m	15 m
PSO [25]	81.30	89.11	93.10
GWO [27]	79.02	85.14	91.30
ABO [15]	77.80	81.61	87.93
BOA [16]	78.27	85.04	91.20
DSA [29]	88.38	91.51	95.42
HPSBA [51]	89.57	95.77	98.66
FBA [19]	90.40	96.24	98.97
BBO (our)	91.67	97.15	99.38

## Data Availability

Not applicable.

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
