# Peer review of "Joint Light-Sensitive Balanced Butterfly Optimizer for Solving the NLO and NCO Problems of WSN for Environmental Monitoring"

_biomimetics, 2023, doi:10.3390/biomimetics8050393_

Round 1

Reviewer 1 Report

The paper is well written. The main question addressed by the research is light-sensitive balanced butterfly optimizer for solving the NLO and NCO problems. The topic is original and relevant in the field. It addresses a specific gap in the field. Light-sensitive balanced butterfly optimizer is used for environmental monitoring. Authors should consider some improvements regarding the delay performance. No further controls are needed. The conclusions are consistent with the evidence and arguments presented, and they address the main question. The references are appropriate.

I have some recommendations.

Related works may be summarized in a table.

Some papers may be referenced as below.

https://doi.org/10.1016/j.compag.2020.105461

https://doi.org/10.1007/s11277-021-08990-3

Minor editing of English language required.

Author Response

***************************************************************************

We appreciate the reviewer for his/her precious and valuable comments that help us improve the quality of the paper. In the revised paper and the response letter, the mentioned issues have been fully considered and addressed.

***************************************************************************

Comments and Suggestions for Authors

The paper is well written. The main question addressed by the research is light-sensitive balanced butterfly optimizer for solving the NLO and NCO problems. The topic is original and relevant in the field. It addresses a specific gap in the field. Light-sensitive balanced butterfly optimizer is used for environmental monitoring. Authors should consider some improvements regarding the delay performance. No further controls are needed. The conclusions are consistent with the evidence and arguments presented, and they address the main question. The references are appropriate.

 Response:

Thank you for your recognition of our paper.

I have some recommendations.

Related works may be summarized in a table.

Response:

We have added a table to summarize the related works in the revised manuscript.

Table1

Author

Problem

Algorithm

Title

Shi et al.

Node localization optimization (NLO)

Particle swarm optimization (PSO)

An improved DV-Hop scheme based on path matching and particle swarm optimization algorithm

Han et al.

Node localization optimization (NLO)

DE algorithm

Enhancing the sensor node localization algorithm based on improved DV-hop and DE algorithms in wireless sensor networks

Zhang et al.

Node localization optimization (NLO)

Enhanced sparrow search algorithm

Multi-Strategy Improved Sparrow Search Algorithm for Solving the Node Localization Problem in Heterogeneous Wireless Sensor Networks

Ghafour et al.

Node localization optimization (NLO)

Enhanced squirrel search algorithm

Improved DV-Hop based on Squirrel search algorithm for localization in wireless sensor networks

Yang et al.

Node coverage optimization (NCO)

Improved Firefly algorithm (FA)

Deploying charging nodes in wireless rechargeable sensor networks based on improved firefly algorithm

Wang et al.

Node coverage optimization (NCO)

Resampled PSO algorithm

Coverage Control of Sensor Networks in IoT Based on RPSO

Miao et al.

Node coverage optimization (NCO)

GWO-EH algorithm

Grey wolf optimizer with an enhanced hierarchy and its application to the wireless sensor network coverage optimization problem

Wang et al.

Node coverage optimization (NCO)

Wolf pack algorithm (WPA)

A novel topology optimization of coverage-oriented strategy for wireless sensor networks

Zhang et al.

Node coverage optimization (NCO)

hybrid HPSBA

HPSBA: A Modified Hybrid Framework with Convergence Analysis for Solving Wireless Sensor Network Coverage Optimization Problem

Some papers may be referenced as below.

https://doi.org/10.1016/j.compag.2020.105461

https://doi.org/10.1007/s11277-021-08990-3

Response:

We have added the references in the revised manuscript as follows:

[1] Tay, M., & Senturk, A. (2022). A new energy-aware cluster head selection algorithm for wireless sensor networks. Wireless Personal Communications, 122(3), 2235-2251.

[2] Bayrakdar, M. E. (2020). Enhancing sensor network sustainability with fuzzy logic based node placement approach for agricultural monitoring. Computers and Electronics in Agriculture, 174, 105461.

Reviewer 2 Report

They claimed that the previous algorithms applied to node
localization optimization (NLO) and node coverage optimization (NCO) problems had low accuracy, and their proposed methods achieved better performance. The related work section is, in my view, still too confusing. I consider that an interesting and attractive method to write a section on the state of the art is to present an advancement in scientific approaches. The authors cited several studies, and I found the entire process to be somewhat confusing. Please revised the related work section. Also, add a related work comparison table that presents the limitations of current work and state-of-the art work. To move the paper toward comparative analysis, the abstract and conclusion should be updated.

.

Author Response

***************************************************************************

Thanks a lot for the invaluable comments and suggestions, these suggestions are very valuable for our paper, and we carefully revise the manuscript according to these suggestions.

***************************************************************************

Comments and Suggestions for Authors

They claimed that the previous algorithms applied to node localization optimization (NLO) and node coverage optimization (NCO) problems had low accuracy, and their proposed methods achieved better performance. The related work section is, in my view, still too confusing. I consider that an interesting and attractive method to write a section on the state of the art is to present an advancement in scientific approaches. The authors cited several studies, and I found the entire process to be somewhat confusing. Please revised the related work section. Also, add a related work comparison table that presents the limitations of current work and state-of-the art work. To move the paper toward comparative analysis, the abstract and conclusion should be updated. 

Response:

As suggested, we have added a table to summarize the related works. In addition, we have revised the “Abstract” and added some discussion in the “Conclusion” Section of the revised manuscript. They are as follows:

Table1

Author

Problem

Algorithm

Title

Shi et al.

Node localization optimization (NLO)

Particle swarm optimization (PSO)

An improved DV-Hop scheme based on path matching and particle swarm optimization algorithm

Han et al.

Node localization optimization (NLO)

DE algorithm

Enhancing the sensor node localization algorithm based on improved DV-hop and DE algorithms in wireless sensor networks

Zhang et al.

Node localization optimization (NLO)

Enhanced sparrow search algorithm

Multi-Strategy Improved Sparrow Search Algorithm for Solving the Node Localization Problem in Heterogeneous Wireless Sensor Networks

Ghafour et al.

Node localization optimization (NLO)

Enhanced squirrel search algorithm

Improved DV-Hop based on Squirrel search algorithm for localization in wireless sensor networks

Yang et al.

Node coverage optimization (NCO)

Improved Firefly algorithm (FA)

Deploying charging nodes in wireless rechargeable sensor networks based on improved firefly algorithm

Wang et al.

Node coverage optimization (NCO)

Resampled PSO algorithm

Coverage Control of Sensor Networks in IoT Based on RPSO

Miao et al.

Node coverage optimization (NCO)

GWO-EH algorithm

Grey wolf optimizer with an enhanced hierarchy and its application to the wireless sensor network coverage optimization problem

Wang et al.

Node coverage optimization (NCO)

Wolf pack algorithm (WPA)

A novel topology optimization of coverage-oriented strategy for wireless sensor networks

Zhang et al.

Node coverage optimization (NCO)

hybrid HPSBA

HPSBA: A Modified Hybrid Framework with Convergence Analysis for Solving Wireless Sensor Network Coverage Optimization Problem

A novel BBO is proposed inspired by the butterflies in nature, which have not only smell-sensitive characteristics but also light-sensitive characteristics. We take the smell-sensitive and light-sensitive characteristics into the local and global search strategies of the proposed algorithm, respectively. In addition, the value of an individual's smell-sensitive is generally positive, which is a non-negligible point. The results show that the BBO has better performance in global search capability and stability than other comparison algorithms. In addition, the simulation results indicate that the BBO-DV-Hop localization optimization algorithm proposed in this study has good stability and accuracy in the NLO problem of WSN. Besides, the proposed BBO also has superior performance on the NCO problem. The position of the initial anchor node also needs to be fully considered to ensure the robustness of the network after positioning. In future studies, certain movable nodes can be deployed in unknown nodes to improve the overall life and anti-interference ability of the WSN.

The performance of the BBO also needs to be enhanced for solving high-dimensional optimization problems. It can be studied in-depth, and more effective improvement strategies can be proposed, which can be {\color{red}applied to engineering optimization, IoT, feature selection, etc. Among them, the node deployment, routing, dynamic networking optimization, and other problems of WSN also can be solved by the SI optimization algorithm. In addition, the research of the 3D WSN is also a study hotspot, including the background of 3D space, underwater, complex mountain, and forest fire monitoring, etc.

Reviewer 3 Report

A novel balanced butterfly optimizer is proposed for environmental monitoring using wireless sensor networks with enhanced performance for node localization and coverage optimizations. It is based on smell perception and light perception characteristics of the butterflies. The paper is well written and organized.

There are lot many good works is undertaken in WSN optimization and its localization challenges. The additional reviews can to be added. Some of the sample works are DOI: 10.1109/TITS.2020.2964604, 10.3390/fi13080210, 10.1007/s42235-022-00288-9, 10.1007/s11356-023-27261-1, etc.

Authors should highlight what different strategies to improved global search abilities are taken. In title, insread of ‘Jointly’, a word ‘Joint’ may be thought.

Tables are figures need to be inserted on the pages at the relevance pages where they are needed.

Minor English edits are required.

Author Response

***************************************************************************

Thanks a lot for the invaluable comments and suggestions, these suggestions are very valuable for our paper, and we carefully revise the manuscript according to these suggestions.

***************************************************************************

Comments and Suggestions for Authors

  1. A novel balanced butterfly optimizer is proposed for environmental monitoring using wireless sensor networks with enhanced performance for node localization and coverage optimizations. It is based on smell perception and light perception characteristics of the butterflies. The paper is well written and organized.

 Response:

Thank you for your recognition of our paper.

  1. There are lot many good works is undertaken in WSN optimization and its localization challenges. The additional reviews can to be added. Some of the sample works are DOI: 10.1109/TITS.2020.2964604, 10.3390/fi13080210, 10.1007/s42235-022-00288-9, 10.1007/s11356-023-27261-1, etc.

Response:

We have added the related references to the literature [3-5] of the revised manuscript.

[3] Ghorpade, S. N., Zennaro, M., & Chaudhari, B. S. (2020). GWO model for optimal localization of IoT-enabled sensor nodes in smart parking systems. IEEE Transactions on Intelligent Transportation Systems, 22(2), 1217-1224.

[4] Ghorpade, S., Zennaro, M., & Chaudhari, B. (2021). Survey of localization for internet of things nodes: Approaches, challenges and open issues. Future Internet, 13(8), 210.

[5] Sharma, S., Khodadadi, N., Saha, A. K., Gharehchopogh, F. S., & Mirjalili, S. (2023). Non-dominated sorting advanced butterfly optimization algorithm for multi-objective problems. Journal of Bionic Engineering, 20(2), 819-843.

  1. Authors should highlight what different strategies to improved global search abilities are taken. In title, insread of ‘Jointly’, a word ‘Joint’ may be thought.

 Response:

Thanks for your suggestion, we have changed the “Jointly” as “Joint” of the title of the revised paper.

  1. Tables are figures need to be inserted on the pages at the relevance pages where they are needed.

Response:

As suggested, we have carefully adjusted the Tables and Figures at the relevance pages of the revised manuscript.

Reviewer 4 Report

This paper proposes a balanced butterfly optimizer and verifies the proposed scheme using benchmark functions compared to other algorithms.

The simulation results show that the proposed scheme has superior performance in terms of stability and accuracy.

This paper is well organized and provides practical results to solve the problem. 

My concerns are as follows.

- Please add the full name of SOTA in the Abstract.

- This paper provides the complexity analysis of the proposed scheme which is required for real system implementation. Although the complexity analysis results show that the implementation of the system is practically available, it would be better to compare the complexity with that of conventional schemes. 

- There are too many self-citations by authors. I think that some of them are not required. 

There are many typos and many unnatural expressions. Examples are as follows:

- Besides, The complexity

- Analyses and results

- The coverage and optimization time of nodes showed an increasing trend, and the number of nodes 426

increased by 1.96 percentage points and 0.54 percentage points respectively year-on-year

- For time-consuming,

- etc. 

Proofreading with a double check should be performed before resubmission. 

Author Response

***************************************************************************

Thanks a lot for the invaluable comments and suggestions, these suggestions are very valuable for our paper, and we carefully revise the manuscript according to these suggestions.

***************************************************************************

Comments and Suggestions for Authors

This paper proposes a balanced butterfly optimizer and verifies the proposed scheme using benchmark functions compared to other algorithms.

 Response:

Thank you for your recognition of our paper.

The simulation results show that the proposed scheme has superior performance in terms of stability and accuracy.

 Response:

Thank you for your recognition of our paper.

This paper is well organized and provides practical results to solve the problem.

 Response:

Thank you for your recognition of our paper.

My concerns are as follows.

- Please add the full name of SOTA in the Abstract.

 Response:

We have added the full name of SOTA in the Abstract of the revised manuscript. Notably, “state-of-the-art” is the full name of the “SOTA”.

- This paper provides the complexity analysis of the proposed scheme which is required for real system implementation. Although the complexity analysis results show that the implementation of the system is practically available, it would be better to compare the complexity with that of conventional schemes.

 Response:

The complexity analysis of the proposed algorithm has been in detailed of the revised manuscript.

- There are too many self-citations by authors. I think that some of them are not required.

 Response:

As suggested, we have replaced unnecessary literature and added some relevant literature [1-5] as follows:

[1] Tay, M., & Senturk, A. (2022). A new energy-aware cluster head selection algorithm for wireless sensor networks. Wireless Personal Communications, 122(3), 2235-2251.

[2] Bayrakdar, M. E. (2020). Enhancing sensor network sustainability with fuzzy logic based node placement approach for agricultural monitoring. Computers and Electronics in Agriculture, 174, 105461.

[3] Ghorpade, S. N., Zennaro, M., & Chaudhari, B. S. (2020). GWO model for optimal localization of IoT-enabled sensor nodes in smart parking systems. IEEE Transactions on Intelligent Transportation Systems, 22(2), 1217-1224.

[4] Ghorpade, S., Zennaro, M., & Chaudhari, B. (2021). Survey of localization for internet of things nodes: Approaches, challenges and open issues. Future Internet, 13(8), 210.

[5] Sharma, S., Khodadadi, N., Saha, A. K., Gharehchopogh, F. S., & Mirjalili, S. (2023). Non-dominated sorting advanced butterfly optimization algorithm for multi-objective problems. Journal of Bionic Engineering, 20(2), 819-843.

Comments on the Quality of English Language

There are many typos and many unnatural expressions. Examples are as follows:

- Besides, The complexity

- Analyses and results

- The coverage and optimization time of nodes showed an increasing trend, and the number of nodes increased by 1.96 percentage points and 0.54 percentage points respectively year-on-year

- For time-consuming,

- etc.

Proofreading with a double check should be performed before resubmission.

 Response:

Thanks for your comments, we have corrected the typos and carefully checked the rest of the revised manuscript.

Reviewer 5 Report

Review of Article #2528190

Title of Article: Jointly light-sensitive balanced butterfly optimizer for solving the NLO and NCO problems of WSN for environmental monitoring

General:

-            You should explain each abbreviation meaning before its primary usage.

-            Avoid too long phrases employing lots of abbreviations, it will decrease understanding.

The conclusion section should emphasize more the utility of the proposed algorithm, with some practical examples.

No comments

Author Response

***************************************************************************

Thanks a lot for the invaluable comments and suggestions, these suggestions are very valuable for our paper, and we carefully revise the manuscript according to these suggestions.

***************************************************************************

Comments and Suggestions for Authors

Review of Article #2528190

Title of Article: Jointly light-sensitive balanced butterfly optimizer for solving the NLO and NCO problems of WSN for environmental monitoring

General:

- You should explain each abbreviation meaning before its primary usage.

 Response:

As suggested, we have added the explanation of each abbreviation meaning in the revised manuscript.

-  Avoid too long phrases employing lots of abbreviations, it will decrease understanding.

 Response:

As suggested, we have modified the long phrases employing, and we have used short sentences for explanations to make it easier for the reader to understand in the revised manuscript.

The conclusion section should emphasize more the utility of the proposed algorithm, with some practical examples.

 Response:

Thanks for your suggestions, we have emphasized the utility of the proposed algorithm in the “Conclusion” section of the revised manuscript.

Round 2

Reviewer 2 Report

The authors improved paper to the desired level & addressed all corrections so accepted.

Acceptable